# Social Sustainability and Ulaanbaatar's 'Ger Districts': Access and Mobility Issues and Opportunities

Iqbal Hamiduddin [1], Daniel Fitzpatrick [1,*], Rebekah Plueckhahn [2], Uurtsaikh Sangi [3], Enkhjin Batjargal [3] and Erdenetsogt Sumiyasuren [4]

1   Bartlett School of Planning, University College London, London WC1H 0NN, UK; i.hamiduddin@ucl.ac.uk
2   School of Social and Political Sciences, University of Melbourne, Melbourne, VIC 3010, Australia; rebekah.plueckhahn@unimelb.edu.au
3   GerHub, Ulaanbaatar 14251, Mongolia; uurtsaikh@gerhub.org (U.S.); enkhjin@gerhub.org (E.B.)
4   Public Lab Mongolia, Ulaanbaatar 14191, Mongolia; erdenetsogt.s@publiclabmongolia.org
*   Correspondence: d.fitzpatrick@ucl.ac.uk

**Abstract:** This paper explores the concept of social sustainability in Ulaanbaatar's ger districts in relation to access and mobility. Although ger districts are well-established in Mongolian culture as ephemeral encampments with transient residents, contemporary ger districts have become large and permanent residential districts that are now home to an estimated one-third of the country's population. The more recent growth of the ger districts has taken place in three decades since Mongolia embraced market-based liberal economics, coinciding with waves of socially and economically-motivated urbanisation. More recently, difficult environmental conditions in rural Mongolia have created new waves of migration. The unfolding situation means that the ger districts have grown with little of the forward planning present in other built areas of the city. In turn, this has led to significant imbalances in the provision of transport services into the ger districts and the problems of access and mobility that this paper has highlighted. This paper has identified community-based local transport and delivery services as one potential means for addressing existing access and mobility shortcomings. Such approaches could provide temporary or complementary services alongside other public policy approaches.

**Keywords:** social sustainability; ger districts; access; mobility; Ulaanbaatar

## 1. Introduction

In common with many middle- and lower-income nations, Mongolia continues to experience high levels of rural to urban migration. In Ulaanbaatar, Mongolia's capital city, many rural migrants move to the 'ger districts' [1,2]—largely unplanned, predominantly residential areas—where many residents live in collapsible felt dwellings known as 'ger' and commonly used by Mongolian mobile pastoralists. Although the concept of urban ger districts is long-established in Mongolia [3] (the older districts having historically begun as seasonal camps around monasteries), newer ger districts in Ulaanbaatar have helped the city to cope with the rapid influx of incomers over the past three decades since Mongolia has embraced economic liberalisation [4]. Today, the ger districts are estimated to house more than half of the capital's population of 1.5m residents [5].

The ger districts are sometimes compared to informal housing areas across the Global South [6,7], and there are some similarities with regards to the provision of basic infrastructure and services. For example, most ger district households are not connected to municipal sewerage and sanitation networks, and houses, known as *baishin*, are not connected to centrally provided heating networks [8]. Most homes feature outside pit latrines and residents are usually obliged to collect clean water from communal water kiosks for a small charge [5]. Communication routes within the ger districts are commonly in the form of unsurveyed and unpaved dirt tracks and paths that have developed organically with the

growth of the districts. In common with newer informal housing areas across the Global South, newer land plots within urban sub-districts utilise marginal sites on hillsides above the formally planned and developed areas of the city. Although the hillsides are often not particularly steep, the combination of topography, poor road quality and constraints to carriageway width and road geometry imposed by patterns of land ownership and built form makes many roads unsuitable for larger vehicles (including public buses). Consequently, transport services are often limited to smaller private cars, taxis and informally operated share-taxis and minibuses [9].

The ger districts also differ from other Global South informal settlements in some important and distinctive ways. Crucially, many land plots are legally recognised, either as owned outright or as a situation where occupiers are granted a form of temporary possession that can last a number of years [6]. A land law introduced in 2002 gives each registered Ulaanbaatar resident the right to a 700sqm plot of land within the greater Ulaanbaatar region [10], providing sufficient space for a fenced land plot ('khashaa'), often large enough to accommodate several family members. In turn, the generous size of the plots means that the overall housing density is relatively low, resulting in a geographically sparse distribution of shops, services and community infrastructure, as well as lengthy travel distances to reach these. Population density therefore presents a further barrier to the development of public transport services. This means that many ger district residents face potential problems of disconnection and 'transport disadvantage', leading, in turn, to the possibility of transport-related social exclusion [10] and transport poverty [11]. Although reliable data are scant, available studies suggest that first journey stage—of being able to access public transport from the home—is a considerable barrier in itself because of the lack of available transport within the ger districts. A second problem is the high level of traffic congestion on major roads (including in the ger districts) that leads to lengthy journeys and journey time unreliability.

Viewed through the lens of social sustainability, the ger districts appear to feature a distinctive set of problems in relation to constrained access and mobility. Using Bramley et al.'s [12,13] basic organising framework of social equity and sustainability of community (here we use 'community' as a geographical descriptor for residents of a given locality), it might be assumed that ger district residents suffer a fundamental inequity in having poor access to public transport [9] that may, in turn, lead to difficulties in sustaining livelihoods, relationships and, in particular, a basic standard of living for vulnerable groups across ger districts. However, this hypothesis has not yet been investigated empirically. This paper therefore aims to make a first attempt to address this shortcoming by examining the impact of existing access and mobility constraints on ger district residents in relation to a set of social sustainability indicators. Secondly, the paper showcases some of the key coping strategies employed by residents to bridge existing access and mobility shortcomings. Lastly, the paper reflects on how significant, continued shortfalls in access and mobility might be addressed through sub-district level, community-based mobility interventions. The latter point is significant because the magnitude of the Ulaanbaatar's ger districts, which are home to approximately one third of Mongolia's entire population, presents a significant barrier to official policy action, given the public resources available.

Social sustainability remains a 'work in progress' [14,15]. Shirazi and Keivani [15] acknowledge a foundational problem that 'there is no single evaluation framework applicable to all disciplines and scales' for evaluating social sustainability in relation to qualities of the built environment. In relation to the disciplinary issue, and the potential for operational frameworks to try to cover an enormous range of different contributory social processes, the earlier work of Bramley et al. [12] and Dempsey et al. [16,17] can be regarded as seminal in providing a robust framing of built environment-influenced social sustainability interactions in relation to two core principles of social equity and sustainability of community. In relation to the issue of scale, a number of authors [17–19] have argued for the importance of the urban neighbourhood as a conceptual unit of evaluation, as it retains a significance for the provision of community infrastructure, amenities and services, as well as social

relations. Larimian and Sadeghi's [20] definition of a 'socially sustainable neighbourhood' provides a useful starting point for operationalising local social sustainability, namely one that: provides residents with equitable access to facilities, services, and affordable housing; creates a viable and safe environment for interaction and participation in community activities; and promotes a sense of satisfaction and pride in the neighbourhood in a way that ensures people would like to live there now and in the future.

A number of built environment-oriented social sustainability evaluation frameworks have emerged in recent years [19–21]. In relation to access and mobility issues specifically, Dempsey et al.'s [17] earlier framing of social sustainability in relation to the two pillars of social equity and sustainability of community framework brings together the discernible outcomes of social equity with the social processes of community sustainability. The social equity pillar focuses directly on aspects of accessibility and mobility to amenities, opportunities and people, such as through the distribution of amenities and opportunities, including employment, as well as the means to access—through transport or other communication forms [17]. The sustainability of community aspects also influences resident access and mobility (albeit less directly) through measures of neighbourhood safety and security—real or perceived—as influenced by local social networks and social capital [11]. Building on the earlier work of Hamiduddin [18], Hamiduddin and Adelfio [22] developed the Dempsey et al.'s two pillar framework into an operational set of indicators (Table 1).

**Table 1.** Dempsey et al.'s social sustainability framework.

| Factor | Indicator |
|---|---|
| Social Equity | Public transport provision |
| | Employment opportunities |
| | Retail offer |
| | Education, health and community infrastructure |
| | Green and recreation spaces |
| | Neighbourhood accessibility measures |
| Community Sustainability | Social interaction |
| | Community participation |
| | Community stability |
| | Pride/sense of place |
| | Safety and security |
| | Demographic profile, relative to wider locale. |
| | Housing access and affordability |

As the focus of this paper is on social sustainability in relation to access and mobility specifically, the list of community sustainability indicators has been tailored for this purpose, with those identified as largely irrelevant removed (Table 2). The availability of public transport services and the ability of residents to access services is a fundamental contributor to social equity [22], particularly in a Global South city such as Ulaanbaatar where a majority of ger district households do not have reliable access to a private car [23–25]. Cordoba et al. [26] note the existence of an historic inverse correlation between mobility and poverty, and Matsuyuki et al. [27] suggest that increased mobility can, in many instances, be taken to be as a proxy for reduced levels of poverty. Access to public transport therefore provides a foundational means to offer mobility, and thus, by definition, access to the opportunities of the city. On the other hand, accessibility can also be provided by integrating services, amenities and employment opportunities with residential spaces [28]. In the case of the ger districts, several local employment opportunities exist by the location of public offices and institutions, including local government, healthcare and education

institutions in each sub-district. Other local employment opportunities are created through small household 'micro enterprises' [29], including local taxi businesses and sole traders. Similarly, the existence of education, primary healthcare and other forms of community infrastructure can reduce a household's 'burden of travel' [30] and promote accessibility across groups that are less mobile. Access to recreational spaces and 'green infrastructure' that are important to health and well-being are also included as a fundamental tenet of social equity. Finally, the degree of local access that different groups experience within a neighbourhood can be influenced considerably by the implementation of measures to promote accessibility and inclusion across a diverse range of groups. Basic measures include the creation of sidewalks to create safe pedestrian or assisted mobility routes [31], the implementation of ramps and the complete removal of physical obstructions to create 'barrier-free' neighbourhoods [32].

**Table 2.** Social sustainability indicators in relation to access and mobility.

| Factor | Indicator | Rationale |
|---|---|---|
| Social Equity | Public transport access | Equity of access to the city; most households do not have reliable access to a private car. |
| | Employment opportunities | Equity of access to basic opportunities to make a living. |
| | Retail offer | Equity of access to the range of basic groceries, pharmacies and other essential provisions. |
| | Education, health and community infrastructure | Equity of access to essential infrastructure, including community services and water. |
| | Green and recreation spaces | Equity of access to opportunities to exercise and interact with the natural environment. |
| | Neighbourhood accessibility measures | Measures implemented to provide neighbourhood accessibility for all residents. |
| Community Sustainability | Social Engagement | Participation in social networks and community processes that develop social capital and enhance well-being and community trust. |
| | Safety and security | Threats and dangers—real or perceived—that can limit the mobility and access of different groups. |

In relation to community sustainability, social engagement means physical access and participation in social networks and social opportunities. Physical access can be regarded as particularly important in global south settings such as the ger districts, where residents may not have reliable access to 'virtual' alternatives to face-to-face contact. Additionally, many familial groups extend across the city, linking districts across familial networks, with some relatives owning apartments, and others owning land plots. Some families work together, utilising different forms of accommodation across familial networks. Furthermore, local place-based social networks can also help to develop local trust within sub-districts that encourages independent mobility among less confident individuals, particularly where transport operators such as taxi drivers live within the community [9]. Built environment factors (such as lighting or the natural surveillance of streets and public spaces) can affect the confidence of different groups to travel across a neighbourhood and on transport services beyond it, at different times of the day. In the case of Ulaanbaatar, as with other global south cities, feral dogs pose an additional threat [9].

## 2. Materials and Methods

The empirical research consisted of two elements of primary data collection. The first data collection element consisted of a household travel survey undertaken in three ger district study sites that provided a range of access and mobility conditions. The second element of data collection consisted of three semi-structured interviews to inform discussion of the sub-district-led response to existing mobility shortcomings through a case study of the recently created taxi union at SKD-31. Here, the findings of a separate survey conducted by GerHub [33] are used to expand upon the qualitative interview material.

The three study sites were selected to represent a range of living conditions, travel connections and terrains found across the ger districts (Figure 1). The three study sites were as follows: 18th Khoroo of Sukhbaatar District, 9th Khoroo of Bayanzurkh District and 31st Khoroo of Songinokhairkhan District. These sites are abbreviated as SBD-18, BZD-9 and SKD-31 throughout the remainder of this paper. SBD-18 and BZD-9 are located in the outer reaches of the city on relatively flat terrain at the base of river valleys. In contrast, SKD-31 is more centrally located but is on steeper hillside terrain, meaning steeper, narrower and less straight access roads that are unsuitable for regular buses.

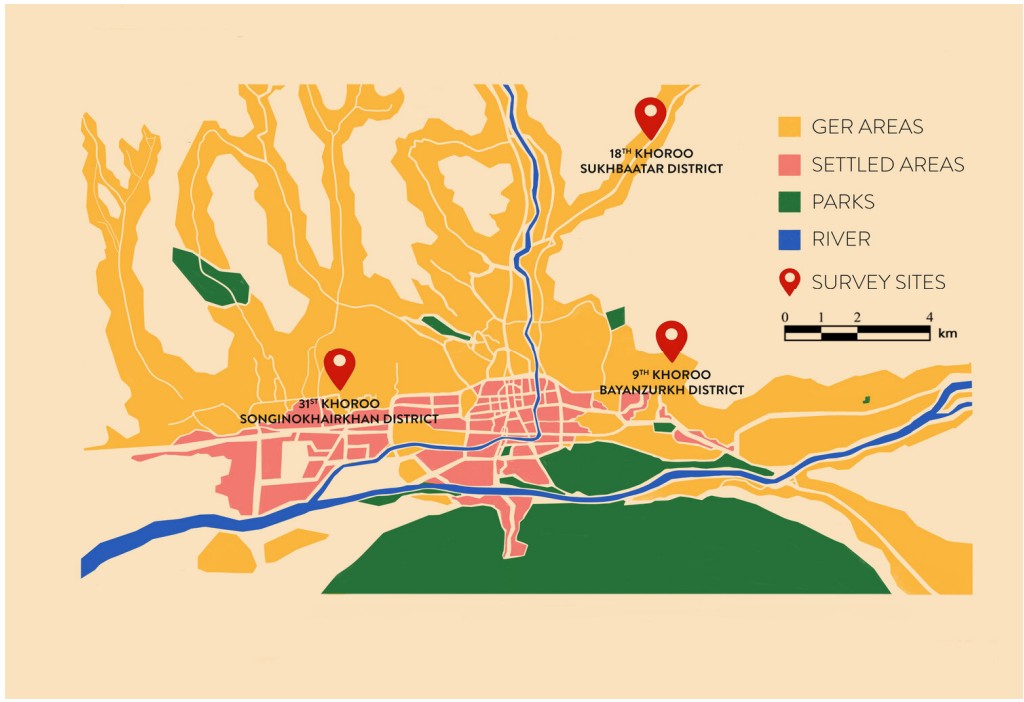

**Figure 1.** Location map showing study sites.

Table 3 summarises a few of the key locational qualities of each the three sites. Despite being more centrally located within the city, SKD-31 is the least connected by public transport, with no direct bus services into the ger district itself. The absence of bus services is related to SKD-31's hillside topography and road width.

**Table 3.** Characteristics of the survey sites.

|  | BZD-9 | SKD-31 | SBD-18 |
| --- | --- | --- | --- |
| Relative Position | Outer city | Inner city | Outer city |
| Topography | Valley floor | Hillside | Valley floor |
| Transport | Bus terminus | No bus | Bus terminus |

The household survey was advertised to residents in each of the three study sites via community Facebook groups existing in each site. Participants were offered a 1500-MNT ($0.5) phone voucher incentive to complete a questionnaire remotely by phone interview due to the onset of the COVID-19 pandemic. This was a change from the originally anticipated strategy of undertaking surveys in-person at each site. Although social media has now been well-established as a means to recruit survey participants, Forgasz et al. [34] identify two highly relevant limitations of this approach; firstly, participants need to have access to the internet, and, secondly, participants must have access to Facebook. Although these are potentially significant constraints to participation in the research, we believe that they would apply more or less equally to residents of each of the three research sites.

A total of 957 travel surveys were collected from across the three study sites from two rounds of data collection. The first survey round was undertaken in March 2020 to capture winter travel patterns, while the second data collection round took place in September 2020 to capture summer-time patterns of life. The travel survey was structured into the following four sections: (i) Personal Characteristics, including time spent at address and car access, (ii) Key Travel Patterns, including a breakdown of journey stages and travel times, (iii) Travel Limitations, including neighbourhood barriers to access and mobility, and (iv) Neighbourhood Life, including social activities and ride-sharing.

Approximately 160 responses were obtained from each of the three sampling sites to provide an overall sample of approximately 480 responses from each survey round, giving an aggregated 95% confidence level with a 4.3% overall margin of error. Broadly speaking, there were a greater proportion of female respondents, although this varied across the three districts (63% in SBD-18, 52% in SGO-31 and 44% in BZD-09). The overall approach was to obtain a minimum of 150 responses from each site during each survey round in order to achieve a 90% confidence level, based on the overall population data available for each of the three sites. However, it should be noted that the continuous and often informal processes of ger district expansion challenges the accuracy of official population data at the sub-district level, which is also difficult to obtain. There was therefore no attempt to match data collection to local demographic data. A further limitation was that data collection took place during a period of disruption because of COVID-19 lockdown restrictions that were in place for some of the time. Therefore, specific aspects of travel data including journey times and costs that need to be treated with a degree of caution and as indicative information only. We therefore present the data as snapshots for each of the sites that together build an overall picture and are individually compared and contrasted in relation to the social sustainability framework set out in this paper. Although this data presents the largest body of evidence on ger district travel currently available in the literature, the limitations to absolute accuracy should be remembered.

## 3. Results

### 3.1. Transport Access

As noted in the Introduction, the ger districts have developed organically through the migration of households from the countryside or elsewhere in Ulaanbaatar with little planning or infrastructure provision. Across Ulaanbaatar, formal bus services are generally restricted to roads prepared in accordance with national highway standards. The ger districts tend to have few of these roads and therefore possess very limited bus services. Within the residential areas of ger districts, residents tend to be reliant on either private hire taxis, shared minivans or '*mickrobus*' that operate on some sealed main roads, or cheaper share taxis that provide an informal transport service. Share taxis tend to operate on an as needed basis along specific routes in that they can often only travel when a car is full in order to maximise income. Away from these kinds of taxi routes, residents without private cars usually have no alternative to walking, using unmade dirt tracks that can become treacherous underfoot during the winter or wet seasons.

The survey data showed that between a third and a half of households across the survey areas owned or had unrestricted access to a private car. However, the distribution of

car ownership and access was skewed towards longer term residents who tended to live in the more established areas of the ger district closer to public transport routes. By contrast, newcomers tended to experience the worst of all circumstances, with homes tending to be located in more peripheral areas away from public transport services and other amenities associated with the core khoroo areas, and also tended to have the lowest levels of private vehicle access.

As Table 4 shows below, SBD-18 and SKD-31 revealed significantly longer overall summer commuting times, across both sexes, compared to winter journey-to-work travel. Longer summertime commute times do not appear to correspond with lengthening journey distances, which are reportedly shorter among residents of two of the three sub-districts during the summer months. This could be due to road construction and maintenance work undertaken during the warmer months, causing traffic congestion on major roads. This is reflected in average overall commuting speeds (average travel speeds have been derived from average journey times stated in the question survey, with journey distances calculated by mapping journey details using ArcGIS), which range from 14.4 km/h among SBD-18 residents during the wintertime to just 4.9 km/h experienced by SKD-31 residents during the summer. Despite being the most centrally located of the study sites within the city with the shortest overall commuting distances, SKD-31 residents have the slowest journeys, reflecting its poor transport connections.

**Table 4.** Journey-to-work characteristics across the three survey sites.

|  | **BZD-9** | **SKD-31** | **SBD-18** |
|---|---|---|---|
| Ave commute time, summer | 56 min | 63 min | 56 min |
| Ave commute distance, summer | 6.5 km | 5.1 km | 9.9 k m |
| Ave commute speed, summer | 7 km/h | 4.9 km/h | 10.6 km/h |
| Ave commute time, winter | 56 min | 47 min | 35 min |
| Ave commute distance, winter | 8.2 km | 6.6 km | 8.4 km |
| Ave commute speed, winter | 8.8 km/h | 8.4 km/h | 14.4 km/h |

To summarise, the travel data in this section confirmed that despite being the most centrally located of the three study sites, SKD-31 was the most disconnected because of the absence of public transport services within the residential area. Residents are therefore required to either undertake a lengthy walk or take a share-taxi (often with considerable waiting times) in order to reach bus services outside of the khoroo. In SKD-31, the hillside topography and the built form of the ger district pose significant barriers to the development of road infrastructure capable for regular bus transport.

### 3.2. Employment Opportunities

The three survey districts had varying levels of local employment, which appeared to be related to the densification of commercial development close to transport hubs (Figure 2). BZD-9 and SBD-18 both show high levels of employment clustering around their bus interchanges, in contrast to SKD-31, which has no bus interchange and, in turn, much lower levels of employment density within the khoroo. This pattern indicates the strong structuring influence of Ulaanbaatar's strategic bus network on urban development.

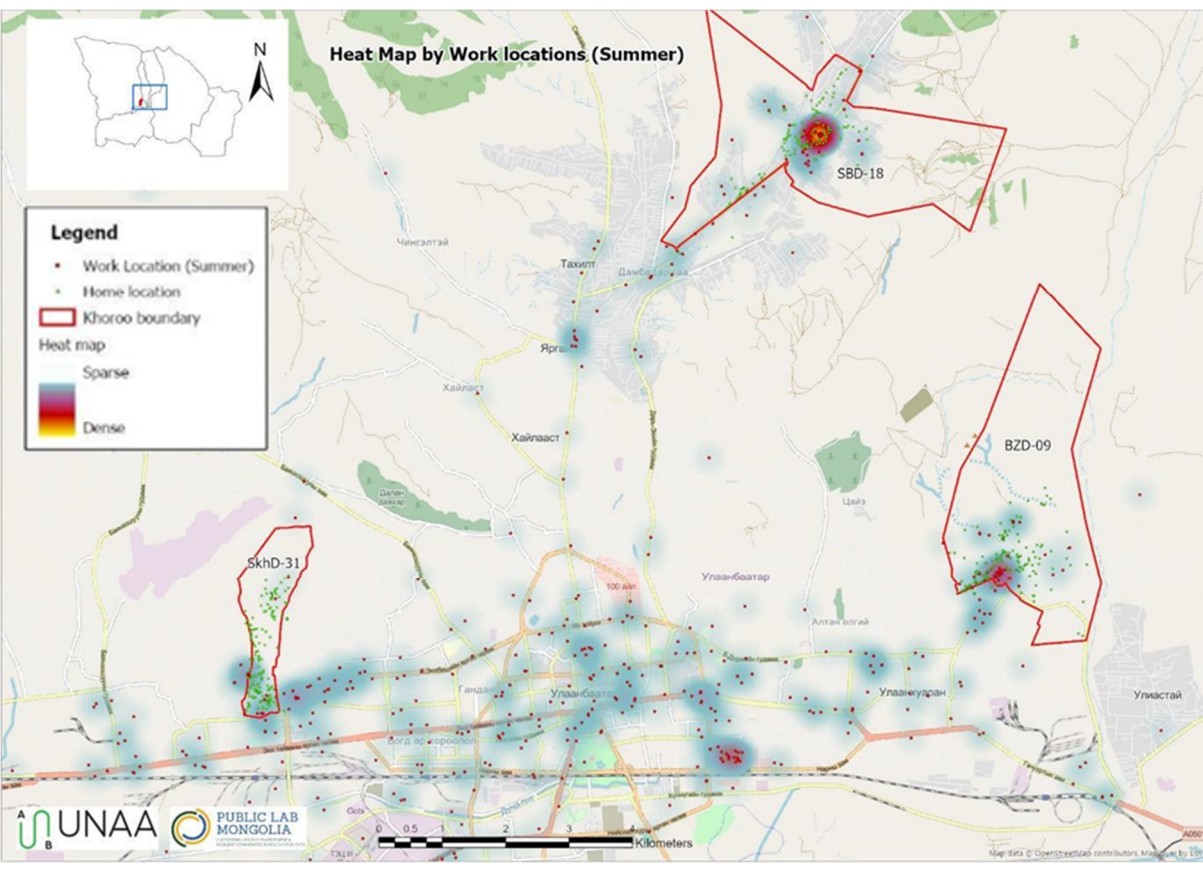

**Figure 2.** 'Heat map' showing spatial distribution of employment locations.

### 3.3. Retail and Services

As retail and services provide major sources of local employment, the spatial disposition is largely as shown for employment in Figure 2, with SBD-18 and BZD-09 both having clusters of commercial activity, including supermarkets, banks, pharmacies and restaurants around their bus terminus. By contrast, SKD-31 has just three small grocery stores distributed across the residential areas. Unsurprisingly, the patterns of use are consistent with local levels of provision, with just 12% of respondents in SBD-18 and 23% in BZD-9 using local shops and services 'rarely' or 'never', compared to 32% in SKD-31 (Table 5).

**Table 5.** Use of local shops and services.

|            | BZD-9 | SKD-31 | SBD-18 |
|------------|-------|--------|--------|
| DAILY      | 11%   | 15%    | 26%    |
| MOST DAYS  | 17%   | 13%    | 25%    |
| SOMETIMES  | 49%   | 40%    | 37%    |
| RARELY     | 22%   | 28%    | 10%    |
| NEVER      | 1%    | 4%     | 2%     |

Once again, the indirect economic impacts of transport services in stimulating commercial activities are very clear in BZD-9 and BZD-18. These retail and service activities are accessible to the wider local population, even if they are not the main focus for household shopping.

### 3.4. Community Infrastructure

Each of the study sites contains essential community infrastructure in the form of nursery and primary education facilities, a primary healthcare facility and a local government office containing a meeting space. Secondary schools are also present in the two larger, outer sub-districts, SBD-18 and BZD-9, whereas secondary school children in SKD-31 travel to an adjacent sub-district to attend school—usually travelling on foot and necessitating the crossing of a busy highway. The majority of homes in the ger districts are connected to the electricity grid, but most rely on pit latrines and very few are connected to the water main. Instead, households are required to collect water from state-operated water kiosks, where a nominal charge is levied. In the absence of private vehicles, many residents travel by foot with a water bowser or cart for transporting containers (Figure 3), to distances of up to 500 m and along unprepared tracks—a particularly onerous undertaking during the cold winter months.

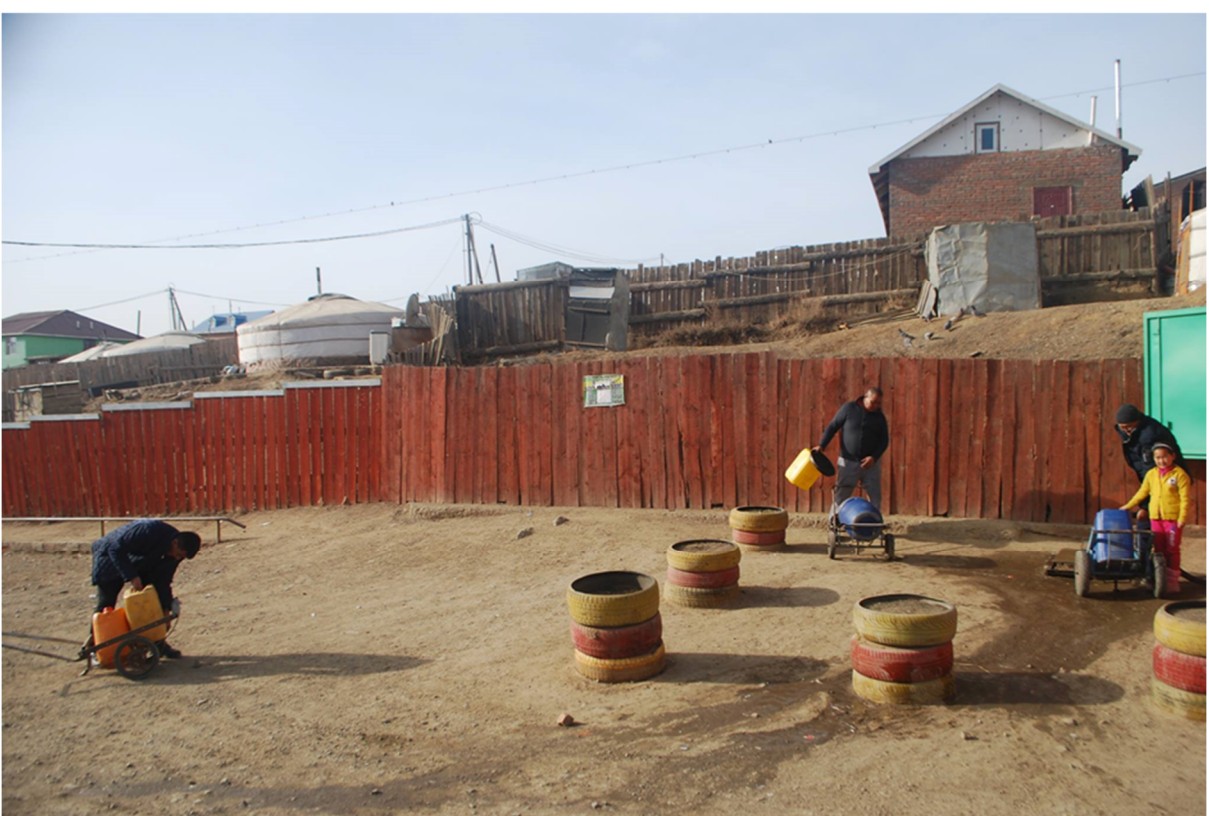

**Figure 3.** Water Collection point in SKD-31.

### 3.5. Green and Recreation Spaces

In contrast to the formally planned apartment areas, where recreation spaces and parks were designed into each 'neighbourhood unit', the ger districts contain very few formally created recreational spaces. In SKD-31, two sports courts and two children's playgrounds have recently been created across the sub-district. SBD-18 and BZD-9 are less well catered-for, with just one public playground area in each sub-district, although in both cases the local secondary schools also have recreational facilities.

### 3.6. Neighbourhood Accessibility Measures

Roads within the ger areas are largely unlit and are often bordered by high "khashaa", or fences, that seal residences off from the street and allow people to secure their property. Poor walking conditions, lighting and the threat of stray animals were cited as contributory factors. Pedestrian walkways are limited in each of the three study sites (particularly

so SKD-31, where there are just two paved roads and only a small section of pavement along each). SBD-18 and BZD-9 feature paved main roads with pedestrian walkways into and around the bus stand at each, but in these sub-districts, conditions for pedestrians deteriorate quickly away from the bus stand, as pedestrians are forced onto unlit and unmade dirt tracks shared by all traffic, and that often bordered by high, blank khashaa fences as illustrated below in Figure 4.

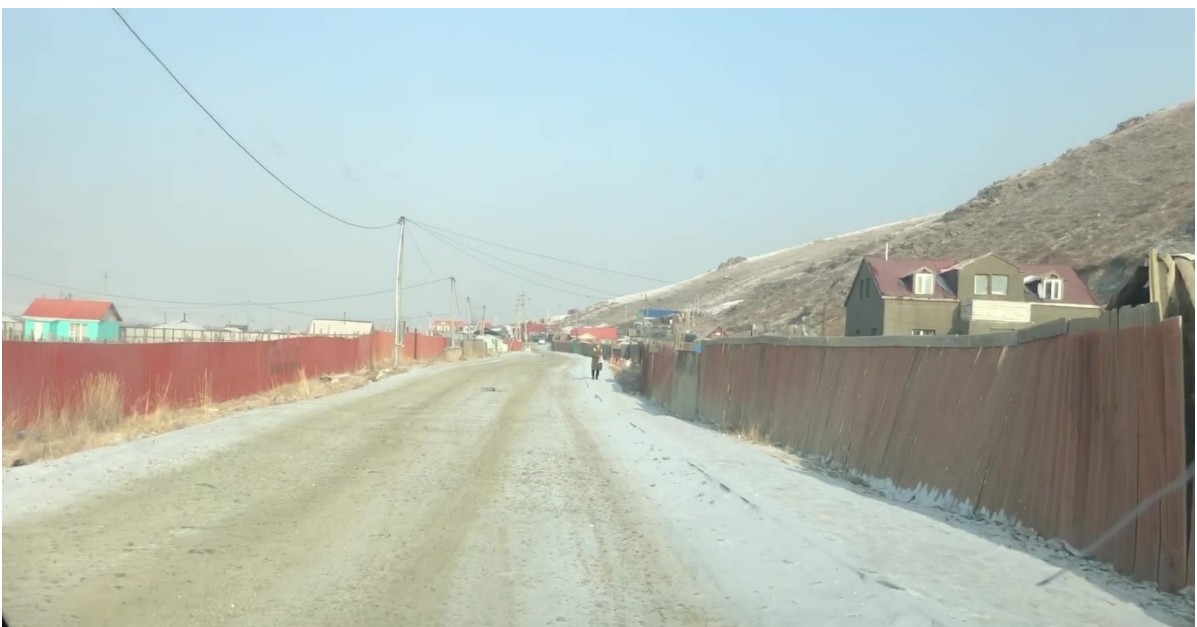

**Figure 4.** Unmade dirt road and pedestrians at BZD-9.

Women often face the additional burden of having to juggle work with childcare responsibilities, including school journeys, entailing lengthy trip-chains. Walking conditions were identified as the greatest barrier in all three sites, relating to the requirement for many residents to travel by foot for local journeys such as to collect water or visit a local grocery store, in all seasons of the year. Poor lighting appeared to be the lesser barrier to local travel across the three sites. However, the scoring range was just 3.6 to 4.5, indicating that all of the factors put forward were problematic to a greater or lesser extent. The findings also indicated SKD-31 to be marginally the most problematic of the three sites for barriers to local travel.

*3.7. Social Engagement*

Access to social networks and engagement in social activity forms an important aspect of community sustainability, in relation to individual and collective well-being and in the development of community trust networks that underpin social capital. Respondents were asked how often they typically had social activity (from a conversation to a planned event) with a neighbour. Responses obtained from BZD-9 and SKD-31 were broadly similar, with relatively low levels of social activity reported (Table 6).

**Table 6.** Reported social activity with neighbours.

|  | BZD-9 | SKD-31 | SBD-18 |
|---|---|---|---|
| DAILY | 5% | 8% | 28% |
| MOST DAYS | 12% | 6% | 9% |
| SOMETIMES | 23% | 26% | 32% |
| RARELY | 40% | 40% | 21% |
| NEVER | 20% | 20% | 10% |

Social activity was markedly higher in SBD-18, with most respondents engaging in some form of interaction with neighbours in a typical week. A similar pattern was found in relation to self-organised ride-sharing (such as for children's school journeys or work commutes), with modest levels of mutual support in SKD-31 (20%) and BZD-9 (23%) eclipsed by a much higher rate of lift sharing in SBD-18 (39%). This is despite the fact that SBD-18 residents enjoyed significantly better access to public transport, with almost all parts of the khoroo located within 800 m or a 10-min walk of a bus stop compared to only half of residents of SKD-31. However, the spatial separation of newcomers and long-term residents was more pronounced among SKD-31 survey respondents compared to the other two districts, and it is the newcomers living in more peripheral areas with lower car access who also potentially experience lower levels of mutual support.

*3.8. Safety and Security*

Away from the taxi routes, residents without private cars usually have no alternative to walk, using unmade dirt tracks that can become treacherous underfoot during the winter or wet seasons.

Table 7 below shows perceived differences in barriers to local travel within the three study areas. A lower score equates to a lower perceived problem. The factors considered were as follows:

- Access to public transport
- Walking conditions, because of the road surface
- Poor neighbourhood lighting
- The threats posed by stray dogs
- Other perceived threats to personal safety

**Table 7.** Perceived barriers to mobility across the neighbourhood (1 = unproblematic/5 = problematic).

| Indicator | Sites | | |
|---|---|---|---|
|  | SBD-18 | SKD-31 | BZD-9 |
| Public Transport | 3.9 | 3.9 | 4.5 |
| Walking Conditions | 4.5 | 4.5 | 4.5 |
| Poor Lighting | 3.8 | 4.1 | 3.7 |
| Stray Dogs | 4.1 | 4.3 | 3.9 |
| Personal Safety | 4.2 | 4.3 | 3.6 |
| (Average Score) | (4.1) | (4.2) | (4) |

Walking conditions were identified as the greatest barrier in all three sites, relating to the requirement for many residents to travel by foot for local journeys (such as to collect water or visit a local grocery store) in all seasons of the year. Poor lighting appeared to be the lesser barrier to local travel across the three sites. However, the scoring range was just 3.6 to 4.5, indicating that all of the factors put forward were problematic to a greater or

lesser extent. The findings also indicated SKD-31 to be marginally the most problematic of the three sites for barriers to local travel.

*3.9. Summary*

Table 8 attempts to draw some initial comparisons between the three sites, in relation to each of the specified access and mobility factors, and particularly to highlight areas of specific concern. The lack of public transport into SKD-31 is problematic because it directly influences residents' access to the wider opportunities of the city and limits economic activities in the sub-district. The lack of public transport reflects the challenging topography and lack of suitable road infrastructure for buses at SKD-31. The table shows that access and mobility across within each of the three study sites is diminished by poor walking conditions, creating an additional challenge for tasks such as water collection, as well as a lack of local public transport within the sub-district.

**Table 8.** Summary of access and mobility conditions at each study site.

| | BZD-9 | SKD-31 | SBD-18 |
|---|---|---|---|
| Public Transport | Bus services to edge terminus Extensive share taxis | No buses Two share taxi routes | Bus services to central terminus Extensive share taxis |
| Employment Opportunities | Economic activity around bus terminus | Limited economic activity | Economic activity around bus terminus |
| Retail + Services | Commercial centre around bus stand 77% regularly use local stores | Sparse offer: three grocery stores 68% regularly use local stores | Commercial centre around bus stand 73% regularly use local stores |
| Community Infrastructure | Nursery, primary/secondary education and primary healthcare Government office/meeting room | Nursery, primary education and primary healthcare Government office/meeting room Community building (GIH) | Nursery, primary/secondary education and primary healthcare Government office/meeting room |
| Recreation | One public sports court, one public play area + secondary school facilities | Two public sports courts, two public play areas | One public sports court, one public play area + secondary school facilities |
| Access | Limited pedestrian routes Public transport access 4.5 Walking conditions 4.3 | Limited pedestrian routes Public transport access 3.9 Walking conditions 4.5 | Limited pedestrian routes Public transport access 3.9 Walking conditions 4.5 |
| Social Engagement | 40% have regular social activity with neighbours 23% organise ride shares | 40% have regular social activity with neighbours 20% organise ride shares | 69% have regular social activity with neighbours 39% organise ride shares |
| Security | Personal safety concerns 3.6 Stray dogs 3.9 Poor lighting 3.8 | Personal safety concerns 4.3 Stray dogs 4.3 Poor lighting 4.1 | Personal safety concerns 4.2 Stray dogs 4.1 Poor lighting 3.8 |

Lastly, there appears to be a relationship between the frequency of social activity between local residents at each study site, and propensity to organise mutual help in the form of ride-sharing, with both regular social activity and organised ride-sharing significantly higher at SKH-18 compared to the other sites. Although the data do not clearly indicate the direction of causality—whether mutual help is the product of social relations, or whether social relations are the outcome of mutual help—it should be noted that 83% of SKH-18 respondents had been resident there for more than five years, compared to 75% at SKD-31 and 70% at BZD-9. It is possible that social relations and propensity to self-organise mutual assistance have been influenced by time lived in their locality.

## 4. Discussion: Improving Mobility through Community Action

The household travel survey findings highlight the problems of travel and access to public transport within each of the three study sites, caused by a poor pedestrian environment and limited local travel options for households without private car access. Hamiduddin and Plueckhahn [9] note that in one ger district a single short journey by share taxi costs 500 Mongolian tögrög (MNT) (approx. $0.2) per person. Relative to the average daily earnings of someone in the ger districts this can be a very substantial cost burden for a single journey stage. How might the situation be improved? Ulaanbaatar's increasing traffic gridlock and poor air quality have put transport on the political agenda. However, given the scale of the task of improving the city's public transport network and the extensive geographic scale of Ulaanbaatar's ger districts, it is unlikely that many areas will receive local improvements soon. However, the ger districts themselves continue to grow with inward migration from the countryside [9]. The question therefore evolves to whether ger district residents improve access and mobility themselves, through community-led action that builds upon existing ride-sharing and social activities evident in the survey data, to a greater or lesser extent, in each of the three study sites.

The entire absence of local public transport and the unreliability and high cost of regular taxis is particularly problematic for residents in Khoroo 31 and 43 of Songinokhairkhan. Responding to these problems, a group of local residents created a private informal taxi service in 2018, establishing a service model that is likely to be of wider interest to residents in other districts across the city. The service provides pick-up and drop-off arrangements to passengers from established stops during the busy times in the morning and evening. This informal service is known to local communities as C176 (named after the route from standard shop to Kindergarten #176). Prior to this service, the community members' only option had been to walk from home to the closest bus station (except for a few residents with private cars). Even though the service was considered expensive by some, it became an essential and widely used by community members. Informal taxis deliver services that are not available through public transportation, but there are issues with their service. For instance, it is up to the drivers to run the service around their own schedules, so some can opt not to work if they consider themselves to have made enough income for the day, leaving the community members with no services. Additionally, there are challenges associated with ensuring road safety, customer service, and varying seasonal conditions of the roads.

The C176 service was initiated by a local resident. He observed these challenges of passengers and taxi drivers and saw the need to organize and formalize the service. Through discussions and consultations with other taxi drivers, he established a group of drivers and took some actions step by step such as registration, issuance of IDs, ensuring that drivers adhere to a set of rules and setting up schedules. The khoroo administration's support during initial phases were valuable, providing meeting space for the group members, allocating taxi stops on the khoroo territory, and installing traffic signs. He introduced the group to the traffic police in the neighbourhood and as a result police knew whom to contact when there was an issue. Passengers saw the advantages of the taxis that belong in the informal taxi group: they considered them more reliable and can even find their belongings if they left it in the taxi, which is a rare feat for the average passenger of an informal taxi. He also established a savings group in November 2015 with a goal to generate sustained income and access to credits in times of need for the participating drivers. Around ten members contributed to the savings around 10,000–30,000 MNT per person daily and started their savings for a fixed term. The savings were used for car repairs or household loans with interest. At the end of the fixed term, the members received their contributed amounts and interest rates were divided among the members. It was considered quite successful and at the end of the first year, some members received 800,000–1 million MNT. The community taxi service became formalised as an NGO in 2020, in order to allow it to expand through the support of other external stakeholders.

During the initial 2020 COVID-19 lockdown period, the C176 provided a food and fuel service to over 20 households to help them through periods of self-isolation. This unintended community delivery service has now been developed into an extended trial, with assistance from the Mongolian NGO Gerhub, to test the longer-term demand and financial sustainability of both the delivery service to households in khoroos 31 and 43, as well as extending passenger transport routes to peripheral areas of the ger district. Market research survey conducted by Gerhub [33] showed a high level of local support for the scheme, with over 80% of respondents stating that they would use the delivery service in the future. Solid (smokeless) fuel would be the single-most popular item for delivery (30%), followed by a combination of solid fuel and water (21%). Comparatively small number of residents would use the service to deliver a combination of groceries and medicine (10%) or just medicine (7%). The findings are interesting. Although water collection is a frequent and onerous task, particularly during the winter months, the travel distances to reach a water kiosk are comparatively short and the kiosks themselves perform the function of a social hub. By contrast, the collection of solid fuel is a less frequent activity, offers little social opportunity, and the travel distances tend to be much greater for many residents. Deliveries of medicine and groceries would probably only be used as a last resort, if residents weren't able to make these trips. Lastly, the Gerhub survey [33] suggested that 65% of residents would use a delivery service 1–2 times per week, with 36% of respondents prepared to pay 1000–2000 MNT, and a further 25% prepared to pay 2000–3000 MNT. By comparison a return trip by share taxi is usually 1000 MNT, and a standard bus fare to the inner city is 1000MNT per leg.

Although the earlier household travel survey data indicated relatively modest levels of self-organised ride-sharing between ger district residents, that range from 20–39% across the three study sites (Table 8), a number of barriers need to be recognised. These include car ownership and availability between community members, organisational factors including travel routes and times, as well as social factors including propensity for mutual assistance and collective self-help outside of family circles. The latter factor is a particular area of uncertainty given how some residents may have recently arrived to urban areas and thus may have limited familial support. The Gerhub survey [33] indicated a strong demand for professionally organised delivery and passenger transport services to better connect residents in more peripheral areas of the ger districts. Such services could provide conditions for improved access to a range of basic goods and services, and with overall access to the city, particularly for those living in more peripheral areas, and for those who are 'time poor'.

## 5. Conclusions

This paper has, for the first time, explored the idea of social sustainability in Ulaanbaatar's ger districts in relation to access and mobility. Although the rapid growth of Ulaanbaatar's ger districts was initially triggered by the new social and economic opportunities presented by Mongolia's switch from communism to a market-based economy in the early 1990s, worsening environmental conditions have more recently added new expansion pressures to the city. The unfolding situation means that the ger districts have grown with little of the forward planning present in other built areas of the city, and this, in turn, has led to significant imbalances in the provision of transport services into the ger districts, and problems of access and mobility that this paper has highlighted. For example, many ger district residents therefore currently spend more than two hours per day travelling to and from work. Journey times are particularly long for a relatively small city of approximately 1.5 m inhabitants, and it reflects both the lack of access to public transport in the ger districts as well as road traffic congestion that inhibits transport services generally across the city. Furthermore, there appear to be seasonal differences in commuting times, which are longer in the summer.

This paper has identified some important structural imbalances in access and mobility. Although the survey data reports that between a third and a half of households across the

survey areas owned or had unrestricted access to a private car, significantly, the distribution of car ownership and access was skewed towards longer term residents who tended to live in the more established areas of the ger district and closer to public transport routes. Furthermore, the survey data showed significant gender differences in car use, with the private car accounting for approximately one third of commuting trips among male respondents compared to under one fifth among females. Thus, with private car travel available to less than a half of households sampled, most journeys out of the khoroo begin with a walk to access public transport, followed by a bus journey, with informal share-taxis generally confined to a limited number of 'trunk' routes. Mapping of employment localities also revealed that districts with public transport terminals had greater levels of local employment, indicating a clustering of economic activity around these transport nodes.

Finally, although the C-176 case study demonstrates the potential for 'organic' community-led approaches to addressing local access and mobility shortcomings in some ger districts, this case poses further questions about whether the model could or should be more widely replicated. Ultimately, these are questions for residents themselves to decide, but the C-176 case does at least show how a community-based mobility scheme could provide temporary or complementary services alongside other public policy approaches. We feel that this scheme is particularly relevant as it cannot be assumed that other strategic public policy initiatives will provide a timely fix to existing mobility and access shortcomings, given the existing scale and continuing growth of the ger districts, against the availability of public resources. We therefore recommend that research on ger district access and mobility be expanded, with due consideration given to the feasibility of community-led approaches to addressing local-scale travel shortcomings.

**Author Contributions:** Conceptualization, I.H., D.F. and R.P.; methodology, I.H., R.P., U.S., E.S.; software, E.S.; validation, U.S. and E.B.; formal analysis, E.S.; investigation, U.S. and E.S.; resources, I.H.; data curation, E.S.; writing—original draft preparation, D.F.; writing—review and editing, I.H. and R.P.; visualization, E.S.; supervision, I.H.; project administration, I.H.; funding acquisition, I.H., R.P., U.S., E.S. and E.B. All authors have read and agreed to the published version of the manuscript.

**Funding:** This research was funded by a Global Challenges Research Fund (GCRF): UCL Internal Small Grant.

**Institutional Review Board Statement:** The study was conducted according to the guidelines of the Declaration of Helsinki, and approved by the Research Ethics Committee of University College London, ID 13911/001 date 26 March 2019 and 13911/002, date 16 June 2020.

**Informed Consent Statement:** Informed consent was obtained from all subjects involved in the study.

**Data Availability Statement:** The data presented in this study are available on request from the corresponding author.

**Conflicts of Interest:** The authors declare no conflict of interest.

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
