# Peer review of "Social Sustainability and Ulaanbaatar’s ‘Ger Districts’: Access and Mobility Issues and Opportunities"

_sustainability, doi:10.3390/su132011470_

Round 1

Reviewer 1 Report

This paper examines access and mobility in Ulaanbaatar. A few questions/comments for the authors: 

Methodology: What are the limitations of the authors' methodological approach? (e.g., online survey recruitment through social media, etc.). Additionally, three semi-structured interviews seems a bit light-weight from the qualitative side. What value does this add in addition to the survey approach? Additional interviews would help enhance this methodological aspect. Finally, what are the implications and/or impacts of conducting this research during the pandemic? The potential impacts of the pandemic on the research findings and any associated limitations should also be discussed. 

The demographics of the sample should also be discussed. How do the survey respondent demographics compare to the general population where the survey was conducted? 

Formatting needs to be checked throughout. Line spacing changes in the conclusion. Some words cut off in tables. 

Author Response

Response to Reviewer Comments: Social Sustainability and Ulaanbaatar’s ‘Ger Districts’: Access and Mobility Issues and Opportunities

We are extremely grateful to the reviewer for engaging with our work in such a detailed and constructive way, especially given the review timeframe. We believe that we have addressed all of the points raised, although we have struggled to control the formatting in the journal template – something that we will liaise with the journal editorial team to correct.  

Here is an account of the revisions that we have made (in red), in relation to each comment.

This paper examines access and mobility in Ulaanbaatar. A few questions/comments for the authors: 

Methodology: What are the limitations of the authors' methodological approach? (e.g., online survey recruitment through social media, etc.).

We have significantly expanded the Methodology section of the paper to include an expanded critique of our data collection approach and limitations, including an additional reference on using social media to recruit research participants[34]

Additionally, three semi-structured interviews seems a bit light-weight from the qualitative side. What value does this add in addition to the survey approach? Additional interviews would help enhance this methodological aspect.

Although it may seem a bit unbalanced, the three interviewees were used to construct a single narrative case study on the community-led taxi service, which we anted to include in this paper as it moves the discussion along from simply reporting and reflecting on the survey findings to the question of ‘what can be done?’. The three interviews are also augmented by data from a separate survey conducted by one of the research partners [33]. We don’t think that this was very clearly stated in the original draft, so the text has been altered and expanded to strengthen this.

Finally, what are the implications and/or impacts of conducting this research during the pandemic? The potential impacts of the pandemic on the research findings and any associated limitations should also be discussed. 

We included some very brief comments in the original draft, but have expanded upon these in the discussion of the limitations of the research.

The demographics of the sample should also be discussed. How do the survey respondent demographics compare to the general population where the survey was conducted? 

We have now included comments on the gender balance of the survey sample. However, although we collected a range of respondent demographic data in our survey, demographic data at the sub-district level is not available so we cannot compare our sample to the sub-district population with any accuracy. We have therefore stated this limitation and tried to reinforce the message that the findings provide snapshots of each district that can be compared to each other, identify access and mobility issues, and provide overall impressions, but fall short of being an ‘absolute’ characterisation of access and mobility for each sub-district.

Formatting needs to be checked throughout. Line spacing changes in the conclusion. Some words cut off in tables. 

Thanks. We have corrected the formatting as much as we can but have struggled with the Word template for this journal (and are not alone with this). We will take this up with the editorial team.  

Reviewer 2 Report

Dear authors,

Submitted manuscript brings research about access and mobility issues and opportunities in Ulaanbaatar’s ‘Ger Districts’. I found the manuscript interesting and useful for transport oriented researchers. However, I can not find it original for now, beacuse of some doubts connected with cited references (later about it). Key arguments are clear, terminology used is appropriate.

Structure of the text is clear and understandable, with optimal extent of chapters. Submitted manuscript is written in high quality English language, but some minor spell check and style improvement should be done.

However, I have some serious remarks regarding the manuscript.

I am concerned about cited references. Namely, I found 4 cited references from Hamiduddin, I. (1st author) and 3 cited references from Plueckhahn, R. (3rd author) and 1 joint cited reference from both authors. In total it is 8 cited references out of 31 cited references which is 25 %. I think it is too much for one manuscript, so I am concrned about orgininality of the manuscript.

I also found one problematic reference in the references list (no. 9). I am concerned about it because the title of the paper is very similar and not published yet, but only submitted. I think such referencing is not appropriate.

Next remark is related to the methodology where more detailed explanation of certain parts is missing. Some issues mentioned in the conclusion were not addressed in the text. Also, I think authors should pay highest attention on the organization of the text, even if it is submission of the first version (e.g. there two totally same paragraphs could be found in the text).

Such issues must be avoided and improved.

Please find all my comments and suggestions below:

Lines 65-66

Transport poverty is a function of transport disadvantage and social disadvanatge (e.g. see Lucas, 2012). Since you don't explicitely mention here social disadvantage, I would suggest to add transport disadvantage instead of transport poverty. Or you could put both terms, but with an emphasize that transport disadvantage together in synergy with social disadvanatge will cause transport poverty. Especially because you mention social aspect in next paragraphs. Please consider both options and decide.

Line 166

Later in the text you mention some anaylises but they are not explained in this section. For example, it is not clear how did you collect the data about average commute time and average commute distance (tab. 4., line 232). Later, you mention scoring (line 311), but it is also unclear what is all about. Also, beside general data about scoring, is 1 best or worse? Or 5 is best or worse? Same is at line 352, for example.

You should explain these issues detailed to be more clear with the methodology.

Line180

Please provide a measure of the map, so interested reader could have better view of the size of the researched area. It should be also part of any map.

Lines 187-199

Since you mention gender distribution at line 222, and laterin the text it would be good to mention gender distribution in the survey.

Line 232

I don't understand, you have two times Ave commute time, summer.

Line 250

The legend is small and hard to read.

Line 251

Two times word ''showing'' in the title of the map.

Lines 307-314 and 354-360

Exactly the same sentences. Please pay more attention, check it carefully and correct it!

Lines 492-497

These data were not analyzed in the text. Please check it carefully and correct it!

Lines 539-540

I found one unappropriate reference: Hamiduddin, I. & Plueckhahn, R. From the Ger Districts to the CBD: contrasts and ienquities of access and mobility in Ulaanbaatar, Mongolia. Submitted manuscript.

Such paper/reference can not be checked as it is submitted, not even accepted, or published. This reference is very similar to this manuscript, so I have concerns about it.

Author Response

We are extremely grateful to the three reviewers for engaging with our work in such a detailed and constructive way, especially given the review timeframe. We believe that we have addressed all of the points raised, although we have struggled to control the formatting in the journal template – something that we will liaise with the journal editorial team to correct.  

Here is an account of the revisions that we have made:

 I have some serious remarks regarding the manuscript.

I am concerned about cited references. Namely, I found 4 cited references from Hamiduddin, I. (1st author) and 3 cited references from Plueckhahn, R. (3rd author) and 1 joint cited reference from both authors. In total it is 8 cited references out of 31 cited references which is 25 %. I think it is too much for one manuscript, so I am concrned about orgininality of the manuscript.

This comment is understandable, but we would like to reassure the reviewer about the authenticity and originality of this paper. The two authors do indeed build from some of their previous work – Hamiduddin in relation social sustainability / mobility and access issues – hitherto undertaken in a European context. A previous conceptual framework is cited and applied in this research, but that is the extent of the similarity and is quite normal practice for academic work to build over time (the works of Bramley et al and Dempsey et al cited in this research [12,13,16,17] do exactly this). Similarly, Plueckhahn is a social anthropologist who has worked on different issues of development in Mongolia for over a decade and therefore has relevant material that is brought in to contextualise this work, but is not focused on transport and mobility issues.

I also found one problematic reference in the references list (no. 9). I am concerned about it because the title of the paper is very similar and not published yet, but only submitted. I think such referencing is not appropriate.

Again, this is understandable. The submitted article referenced [9] is thematically related, but is a separate piece of work, using an entirely different data set, comparing travel behaviour between residents of one ger district (SKD-31) and residents of an adjacent formally planned and constructed part of the city. This work served as a useful forerunner and pilot for the work reported in this paper but was conducted entirely separately. The paper [9] is now accepted for publication in Local Environment and will be appearing online shortly.The reference has been updated to reflect this recent change – and a DOI will be added as soon as it is available.

Next remark is related to the methodology where more detailed explanation of certain parts is missing. Some issues mentioned in the conclusion were not addressed in the text. Also, I think authors should pay highest attention on the organization of the text, even if it is submission of the first version (e.g. there two totally same paragraphs could be found in the text).

This relates to a comment below

 Please find all my comments and suggestions below:

Lines 65-66

Transport poverty is a function of transport disadvantage and social disadvanatge (e.g. see Lucas, 2012). Since you don't explicitely mention here social disadvantage, I would suggest to add transport disadvantage instead of transport poverty. Or you could put both terms, but with an emphasize that transport disadvantage together in synergy with social disadvanatge will cause transport poverty. Especially because you mention social aspect in next paragraphs. Please consider both options and decide.

 Very useful points – thank you. The text has been altered here with references to recent work by Karen Lucas.

 Line 166

Later in the text you mention some anaylises but they are not explained in this section. For example, it is not clear how did you collect the data about average commute time and average commute distance (tab. 4., line 232). Later, you mention scoring (line 311), but it is also unclear what is all about. Also, beside general data about scoring, is 1 best or worse? Or 5 is best or worse? Same is at line 352, for example.

You should explain these issues detailed to be more clear with the methodology.

 Thank you. The methodology section has now been greatly expanded to provide more details on the data collected and how the analysis was performed. The caption to Table 7 explains the scoring (accidentally removed from a previous draft).

 Line180

Please provide a measure of the map, so interested reader could have better view of the size of the researched area. It should be also part of any map.

 The map now has a scale ruler

 Lines 187-199

Since you mention gender distribution at line 222, and laterin the text it would be good to mention gender distribution in the survey.

 We have now included a note on gender distribution in the Methodology

Line 232

I don't understand, you have two times Ave commute time, summer.

 Should read ‘Winter’ – now corrected

Line 250

The legend is small and hard to read.

 The legend on this map has been expanded

Line 251

Two times word ''showing'' in the title of the map.

 Corrected

Lines 307-314 and 354-360

Exactly the same sentences. Please pay more attention, check it carefully and correct it!

 Corrected

Lines 492-497

These data were not analyzed in the text. Please check it carefully and correct it!

 Corrected

Lines 539-540

I found one unappropriate reference: Hamiduddin, I. & Plueckhahn, R. From the Ger Districts to the CBD: contrasts and ienquities of access and mobility in Ulaanbaatar, Mongolia. Submitted manuscript.

Such paper/reference can not be checked as it is submitted, not even accepted, or published. This reference is very similar to this manuscript, so I have concerns about it.

Please see our comment above

Reviewer 3 Report

The article, drawing on the Ulan Bator ger districts study case, provides a very interesting study about the  fairness issues related to public transport and mobility availability and access to social capabilities on behalf of  people living in transitional urban area.  

The study properly referring to key literature  about mobility nad transport  services availbility and social/community sustainability provides a step further in tailoring fitting indicators to asses the context situation. Jointly the article allow to shed light on socially self-produced and bottom inititatives aimed to cope with the overall shortcoming of the mobility possibilities owed to weak urban structures, public inability to plan and to manage the basic transport services and viable mobility infrastructure.

Even though the sound methodology approach of the article and the originality and of the meaningfulness of the context studied on behalf of the referee a minore revision strive i needed to address and achieve:

  • a better connection between the result and discussion sections. Indeed, discussion section introduces the self-relied set of initiatives undertaken by local community to deal with some mobility/access system shortcomings by adopting a mainly descriptive approach that misses in pointing out which are the key conceptual and operational relationships with the results raised in the previous part. The earned impression for the reader is to face a completely new topic of study;
  • Conclusion section relies still too much on the general matter of the study description, repeated from the introductive section, whereas what it seems lacking are some clear remarks on  key conclusions of the study and the prospect for further research demandsissued by the article. 

Moreover some corrections of the text are needed, specifically:

  • line 114 is lacking of a full stop;
  • table 4 : the 4th string improperly repeat “summer” intead of “Winter”;
  • Safety and security paragraph si quite a repetition of Neighborhood accessibility measures Especially lines 354 to 360 totally repeat lines 307 to 314;
  • Line 406 correct typo “Inormal”;
  • Line 502 does not relate anyhow to data results or discussion section.

Author Response

We are extremely grateful to the reviewer for engaging with our work in such a detailed and constructive way, especially given the review timeframe. We believe that we have addressed all of the points raised, although we have struggled to control the formatting in the journal template – something that we will liaise with the journal editorial team to correct.  

Here is an account of the revisions that we have made:

  • a better connection between the result and discussion sections. Indeed, discussion section introduces the self-relied set of initiatives undertaken by local community to deal with some mobility/access system shortcomings by adopting a mainly descriptive approach that misses in pointing out which are the key conceptual and operational relationships with the results raised in the previous part. The earned impression for the reader is to face a completely new topic of study;

We agree that the connection between the Findings in section 3 and the Discussion in section 4 needed to be strengthened. We have therefore edited paragraphs 1 and 2 in section 4 to provide stronger lines of connection, particularly on the latent potential for community collaboration for addressing mobility shortcomings, which feel is the key line of connection between the two sections.  

  • Conclusion section relies still too much on the general matter of the study description, repeated from the introductive section, whereas what it seems lacking are some clear remarks on  key conclusions of the study and the prospect for further research demands issued by the article. 

Agreed – there is too much repetition of contextual material. We have therefore edited this down (although we have retained mention of recent environmental migration as this is becoming an increasingly important contributor to ger district growth). The Conclusion finishes with suggestions for lines of further research, as suggested by the reviewer. 

Moreover some corrections of the text are needed, specifically:

  • line 114 is lacking of a full stop; - now corrected
  • table 4 : the 4th string improperly repeat “summer” intead of “Winter”; - now corrected
  • Safety and security paragraph si quite a repetition of Neighborhood accessibility measures Especially lines 354 to 360 totally repeat lines 307 to 314; - now corrected (also mentioned by Reviewer 1)
  • Line 406 correct typo “Inormal”;- now corrected
  • Line 502 does not relate anyhow to data results or discussion section. – this comment seems to be in relation to the ger district growth? This section has been edited-down.

Round 2

Reviewer 2 Report

Authors performed all required changes in the manuscript, or explaned adequately problematic issues. 

The only minor remark is to address figures and tables numbering issue. Namely, I think it should be written by numbers, not by words (e.g. Table 1., not Table one, etc.).